# Neural network variational Monte Carlo for positronic chemistry

Gino Cassella [1] ✉, W. M. C. Foulkes [1], David Pfau [1,2] & James S. Spencer[2]

Quantum chemical calculations of the ground-state properties of positron-molecule complexes are challenging. The main difficulty lies in employing an appropriate basis set for representing the coalescence between electrons and a positron. Here, we tackle this problem with the recently developed Fermionic neural network (FermiNet) wavefunction, which does not depend on a basis set. We find that FermiNet produces highly accurate, in some cases state-of-the-art, ground-state energies across a range of atoms and small molecules with a wide variety of qualitatively distinct positron binding characteristics. We calculate the binding energy of the challenging non-polar benzene molecule, finding good agreement with the experimental value, and obtain annihilation rates which compare favourably with those obtained with explicitly correlated Gaussian wavefunctions. Our results demonstrate a generic advantage of neural network wavefunction-based methods and broaden their applicability to systems beyond the standard molecular Hamiltonian.

The positron—the positively charged anti-particle of the electron—was first postulated by Dirac[1] almost a century ago. Today, the once exotic notion of anti-matter is technologically relevant in a variety of fields, such as medical physics[2], astrophysics[3], and materials science[4,5].

Experimental apparatus for trapping large numbers of positrons continues to grow in sophistication, offering a glimpse into a world of exotic antimatter chemistry[6]. These advances motivate the development of improved computational tools able to describe positronic bound states and thus accelerate the continued development of new antimatter-based technologies. In the present work, we make progress in this direction by developing a highly accurate method for the ab initio calculation of the ground-state properties of bound states between positrons and ordinary molecules.

Despite annihilating upon contact with an electron, positrons readily form bound states with ordinary molecules. The formation of these bound states, enabled by an incident positron exciting a vibrational Feshbach resonance which subsequently decays to the vibrational ground state, results in greatly enhanced annihilation rates. This process has recently been successfully exploited in positron annihilation spectroscopy experiments[7–14], where the energy-dependent annihilation rate enables sensitive measurements of the positron binding energy[15].

Theoretical calculations of positron binding energies and annihilation rates have been performed using many of the standard tools of computational chemistry. Positron binding to atoms and molecules has been studied using wavefunction expansions in explicitly correlated Gausians with the stochastic variational method (ECG-SVM)[16–24], configuration interaction (CI) methods[25–31], and quantum Monte Carlo (QMC) methods[32–40]. The annihilation rate and lifetime of positrons in solids, particularly in the presence of defects, has been studied using density functional theory[41,42] and quantum Monte Carlo methods[43,44].

Despite the intense theoretical interest, describing the positronic wavefunction remains challenging for several reasons. As a result of the strong correlations enabled by the attractive interaction between electrons and the positron, molecular bound states often resemble a virtual positronium atom weakly bound to the molecule[45]. Due to the repulsive potential between the nuclei of the host molecule and the positron, these virtual atoms are localized away from the nucleus. Accurately describing such a wavefunction (particularly the electron–positron cusp) in a basis of single-particle orbitals centred on the nuclei requires the inclusion of basis functions with very large angular momenta[46]. This results in very slow convergence of CI calculations with the maximum angular momentum of functions included in the basis. Furthermore, the positronic density is typically highly diffuse, requiring the augmentation of standard basis sets with additional diffuse basis functions[27,29–31].

[1]Dept. of Physics, Imperial College London, London SW7 2AZ, UK. [2]DeepMind, London N1C 4DJ, UK. ✉e-mail: g.cassella20@imperial.ac.uk

The most successful description of positron binding to molecules comes from a recent work that develops a many-body theory of positron binding to molecules and produces positron binding energies in close agreement with experimental measurements[47]. In their work, Hofierka et al. highlight the shortcomings of QMC methods applied to positron binding, particularly the lack of calculations for large non-polar molecules, which constitute the majority of experimentally studied systems[15]. Here, we address this shortcoming.

We propose a new approach to calculating the ground state properties of molecular positronic bound states, based on recently developed neural network wavefunction ansatze for QMC[48]. The Fermionic neural network (FermiNet) models the many-body wavefunction without referencing a set of basis functions. This conveniently sidesteps a number of the aforementioned difficulties in describing positronic wavefunctions. We extend FermiNet to represent the positronic component of the wavefunction on an even footing with the electronic component. With a minimal alteration to the neural network architecture, we obtain a flexible and accurate ansatz for mixed electron–positron wavefunctions. We calculate positron binding energies and annihilation rates for a range of systems with qualitatively distinct mechanisms for positron binding and obtain state-of-the-art accuracy for the ground-state energy in these systems. Our method yields a positron binding energy for benzene in close agreement with the experimental value and the many-body theory of Hofierka et al.[47], and we obtain annihilation rates which compare favourably with highly accurate ECG-SVM calculations for alkali metal atoms and small molecules.

## Results

We benchmark the accuracy of FermiNet in calculations of the ground-state energy, the positron binding energy, and the positron annihilation rate for a series of well-studied positronic systems. These are presented here in (approximate) order of increasing complexity. Unless otherwise specified, all results presented herein were obtained using the network architecture and training protocol detailed by Spencer[49]. We pre-train the electron functions to the Hartree-Fock solution of the bare molecule and do not pre-train the positron functions. Errors in energy expectation values are evaluated using a reblocking approach[50] to account for sequential correlations in the Metropolis-Hastings sampling.

### Binding energies

The positronium (Ps) atom (consisting of a bound electron and positron) and hydrogen form a stable molecule. Here we work within the Born-Oppenheim approximation, neglecting the proton's motion. Almost exact ECG calculations are available for this system[16], yielding a ground-state energy of $E_0 = -0.7891794$ Hartrees. We obtain a ground state energy of $E_0 = -0.789144(3)$ Hartrees.

The first ionization energy of sodium, 0.1886 Hartrees, is smaller than the binding energy of the Ps atom, 0.25 Hartrees. The positronic sodium atom is then more accurately described as a bound complex of a positronium atom and a sodium cation. The binding energy of this complex is calculated as $\epsilon = E([Na^+, Ps]) - E(Na^+) - 0.25$. We fail to predict binding without variance matching, obtaining $\epsilon = -0.37$ milli-Hartrees. Utilizing a variance matching procedure, we obtain $\epsilon = 0.32(15)$ milliHartrees, predicting binding in agreement with previous ECG-FCSVM ($\epsilon_{FCSV\,M} = 0.473$ milliHartrees[51]) calculations.

We obtain a positron binding energy of 0.01618(9) Hartrees for the magnesium atom. This agrees with previous ECG-FCSVM (0.016930 Hartrees[28]) and CI (0.017099 Hartrees[28]) results.

We have calculated the ground-state energy of LiH and its positronic complex for a range of interatomic separations (see Data Tables in Supplementary Material). Fitting these potential energy surfaces using Nesterov's algorithm, implemented in the MOLCAS package, we obtain equilibrium bond distances of 3.0196 Bohr, with ground-state energy -8.07050(1) Hartrees for LiH, and 3.371 Bohr with a ground state energy of -8.10774(1) Hartrees for [LiH, e⁺]. These deviate very slightly from the widely accepted literature values of 3.015 Bohr (LiH) and 3.348 Bohr ([LiH, e⁺]). We have calculated ground-state energies at the canonical separations for comparison with previous results, shown in Table 1. Our calculations are not only in excellent agreement with previous work but are seen to yield the most accurate variational result for [LiH, e⁺].

We have calculated the ground-state wavefunction of BeO over a range of interatomic separations—from below the equilibrium separation to dissociation. We fit the potential energy surfaces using MOLCAS and obtain an equilibrium bond distance of 2.515 Bohr with a ground-state energy of -89.90572(4) Hartrees for BeO, and 2.530 Bohr with a ground-state energy of −89.93082(3) Hartrees for [BeO, e⁺]. We compare our results against previous calculations in Table 1 and find that FermiNet yields the most accurate variational result for [BeO, e⁺].

The electronic ground state of BeO transitions from the spin-singlet configuration at the equilibrium interatomic separation to the spin-triplet configuration at the dissociative limit. We enforce the appropriate ground-state spin configuration by choosing $S_z = 1$ (as FermiNet is a spin-assigned wavefunction) at wide interatomic separations. The resulting potential energy surfaces are plotted in Fig. 1. At ~4 Bohr, the electronic ground-state transitions between the spin-singlet and spin-triplet, causing a sharp change in the ground-state dipole moment and the resulting positron binding energy with the molecular ground state, which vanishes almost completely. At ~6 Bohr, we see a smooth transition between two qualitatively distinct binding modes between the molecule and the positron—binding to the molecular dipole field below this separation and binding exclusively to the lone beryllium atom beyond this separation. This is readily seen by visualizing the ground-state positron density on either side of the

**Table 1 | Ground state energy of LiH and BeO, and their positronic complexes, obtained via various computational methods at equilibrium bond-length**

| Method | LiH | | | BeO | | |
|---|---|---|---|---|---|---|
| | Bare | Positronic | Binding energy | Bare | Positronic | Binding energy |
| FermiNet-VMC | −8.07051(1) | **-8.10775(1)** | 0.03723(2) | **−89.90572(4)** | **−89.93082(3)** | 0.02510(7) |
| SJ-VMC | −8.06307(3)[36] | −8.08034(4)[36] | 0.01727(7) | −89.3173(25)[32] | −89.3365(13)[32] | 0.0192(38) |
| SJ-FN-DMC | −8.070045(38)[36] | −8.10718(11)[36] | 0.03714(15) | −89.7854(13)[32] | −89.8134(12)[32] | 0.02800(25) |
| CISD | −8.03830[25] | −8.05530[25] | 0.017 | – | – | – |
| MRD-CI | −8.06827[27] | −8.09764[27] | 0.02937 | −89.759352[29] | −89.773133[29] | 0.013781 |
| ECG-SVM | **−8.07054**[23] | −8.10747[23] | 0.03693 | – | – | – |
| Hofierka et al.[47] | – | – | 0.039(1)*[47] | – | – | – |

Theoretical (not statistical) uncertainty estimated from the comparison between different levels of theory as described by Hofierka et al., caption of Table I.
Statistical errors are omitted where they are smaller than the reported precision or otherwise omitted in the referenced source. The lowest variational energy in each column is bolded.

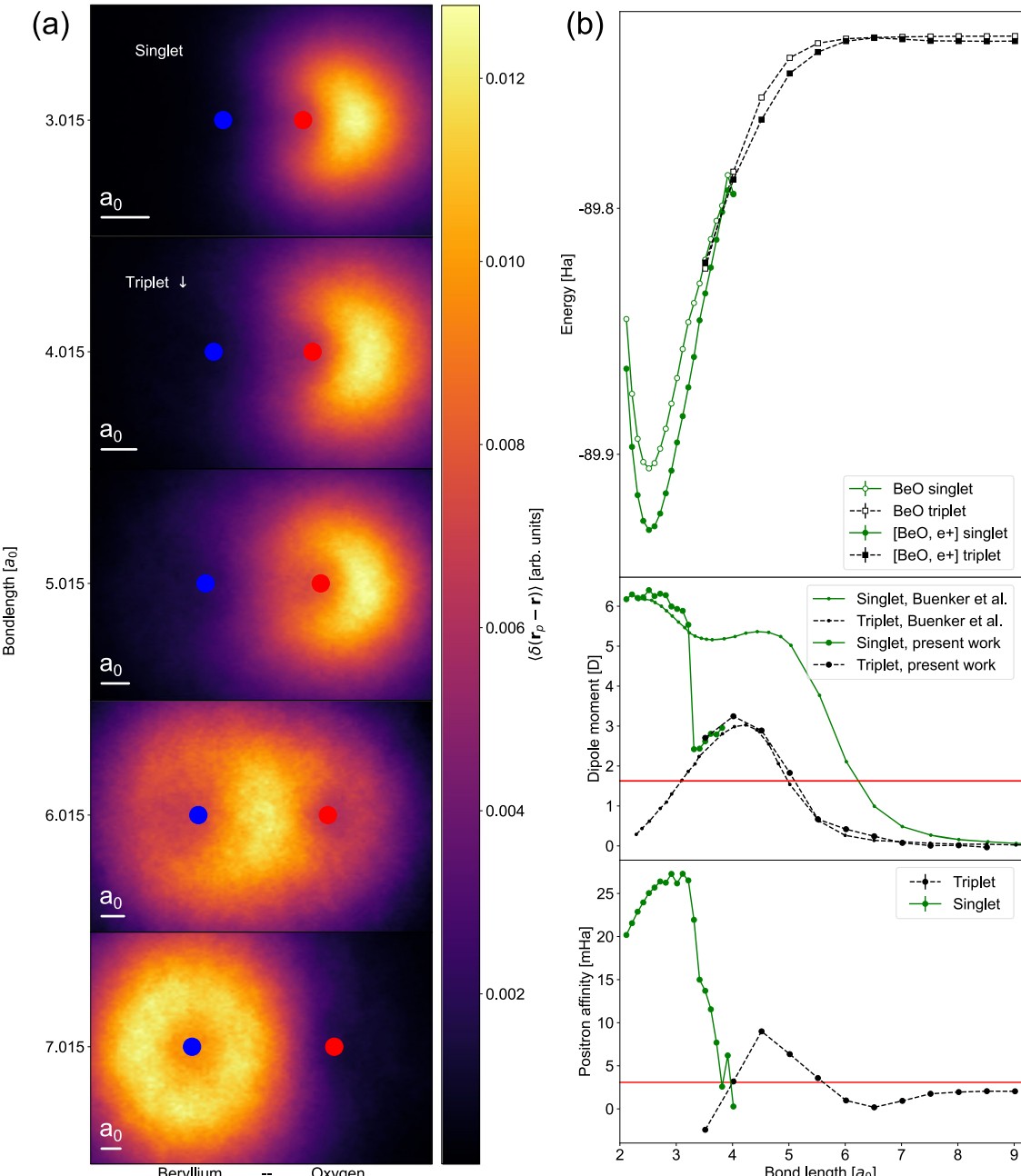

**Fig. 1 | Positron binding characteristics of beryllium oxide. a** Ground-state positronic density, projected into the molecular plane, of a positron attached to a beryllium oxide molecule over a range of interatomic distances, accumulated via MCMC sampling. Image scale has been normalized by bondlength. The scale bar in the bottom left of each panel indicates the relative size of one Bohr radius, $a_0$. The blue (red) marker indicates the position of the beryllium (oxygen) nucleus. **b** Energy (of the bare molecule and positronic complex), dipole moment (of the bare molecule), and positron binding energy of beryllium oxide over a range of interatomic distances. Solid (dashed) lines in the dipole moment and positron binding energy plots indicate values accumulated for the electronic singlet (triplet) projected wavefunction. Error bars indicating standard errors in the Monte Carlo estimate of each quantity are smaller than the markers. Dot markers in the dipole moment plot indicate the corresponding values obtained by Buenker et al.[29]. Horizontal red lines in the dipole moment and positron binding energy plots indicate the mean-field critical value for positron binding to the molecular dipole field and the positron affinity of a lone beryllium atom[68], respectively.

maximum, as shown in Fig. 1 for the triplet state, and corroborated by the dipole moment falling below the critical threshold for binding (~1.625 Debye) at the transition.

For the dilithium molecule, we obtain a positron binding energy of 66.75(2) milliHartrees at an equilibrium bond length of 5.051 Bohr[52]. The molecular binding energy of $Li_2$ from our calculations is -38 milliHartrees, and the positron binding energy of a lone lithium atom is known from the literature to be ~2 milliHartrees[53], meaning this system is very stable against the $[Li_2, e^+] \rightarrow [Li, e^+] + Li$ dissociation channel. The

one-particle density of the positronic ground state of dilithium is shown in Fig. 2a.

The experimental positron binding energy of benzene is 5.51 milliHartrees[15]. Utilizing our variance-matching procedure (described in detail for benzene in the Supplementary Material), we obtain a finite positron binding energy of 4.1(3) milliHartrees. This falls in very close agreement with the binding energy obtained by Hofierka et al. of ~4.26 milliHartrees[47]. The one-particle density of the positronic ground state of benzene is shown in Fig. 2b.

## Annihilation rates

The largest contribution to the annihilation process between positrons and electrons is the two-photon ($2\gamma$) annihilation rate, which is proportional to the positron-electron contact density. This quantity is readily obtained from our calculations. We have calculated $2\gamma$ positron annihilation rates (according to the procedure detailed in the Supplementary Material) for every system studied, listed in Table 2.

Positron annihilation rates are highly sensitive to the accurate description of the coalescence between the positron and electrons. This sensitivity, and the fact that annihilation rates are not a variational quantity, hinders the comparison between annihilation rates calculated by different methods. However, of the other methods compared in Table 2, ECG-SVM captures this feature of the wavefunction best by construction. We see that our FermiNet-VMC annihilation rates are in close agreement with ECG-SVM results for positronium hydride, lithium hydride, and the alkali metal atoms. On this basis, we reason that our approach produces accurate annihilation rates, and therefore offers an accurate description of electron–positron correlations. We note that the many-body theory results of Hofierka et al. appear to overestimate the annihilation rate in lithium hydride compared to ECG-SVM, suggesting that FermiNet-VMC may be more suitable for the calculation of annihilation rates.

## Discussion

The selection of systems presented here spans a broad range of positron binding phenomena: positronium formation, binding with an induced atomic dipole moment, binding with a static molecular dipole moment, and binding due to correlations with covalent bonding electrons in molecules. In all cases where benchmarks are available, FermiNet-VMC produces excellent and, in some cases, state-of-the-art results for the positron binding energy. From the density plots presented, it is clear that this performance is consistent between wavefunctions with very different qualitative characteristics. An identical ansatz is used for every system studied–we have not employed any fine-tuning or system-specific treatments. An important aspect of our approach is that, due to being a basis-set-free method, we do not provide any information about the nature of the positron binding in the input to the calculation (e.g., via the placement of basis set functions). Rather, the location and nature of the positron binding emerges naturally during optimization of the wavefunction. The high level of accuracy achieved across various systems shows that FermiNet offers a flexible and accurate ansatz for mixed electron–positron wavefunctions.

To date, quantum Monte Carlo calculations of positron binding have focused on small, polar molecules. As pointed out by Hofierka et al.[47], accurate quantum Monte Carlo results for large non-polar organics, which comprise the majority of experimentally relevant systems, are lacking. We believe that the present work fills this gap. Our results for positronium hydride, sodium and magnesium atoms, and small diatomic molecules demonstrate that our approach can achieve state-of-the-art accuracy compared with previous work. Further, our results for the non-polar dilithium and benzene molecules demonstrate that this accuracy is retained when describing modes of positron binding governed entirely by strong electron–positron correlation effects. The results in Fig. 2 offer an intuitive understanding of the binding mechanism between non-polar molecules and positrons: correlation-dominated binding is facilitated by the presence of a centre of increased electronic density away from the atomic nuclei of a molecule. In dilithium, this is the covalent bond, and in benzene, this is the increased electronic density in the centre of the molecule from the delocalization of the $\pi$-bonds in the ring.

Our result for the positron binding energy of benzene fall within chemical accuracy (-1.6 milliHartrees) of the experimental value, but this level of accuracy is insufficient for other species of experimental interest. Many chemical species possess positron binding energies below chemical accuracy (see, e.g., the experimental binding energies in ref. 11, where 15 small organic molecules are found to possess binding energies below chemical accuracy), and obtaining this level of accuracy for relative energies represents a significant challenge to computational chemistry methods. We believe that the necessary improvement in accuracy, and obviation of our need to utilize variance matching, can be achieved by adopting the recently introduced Psi-Former architecture[54], an improvement upon the FermiNet architecture utilizing Transformer networks[55], given that the PsiFormer

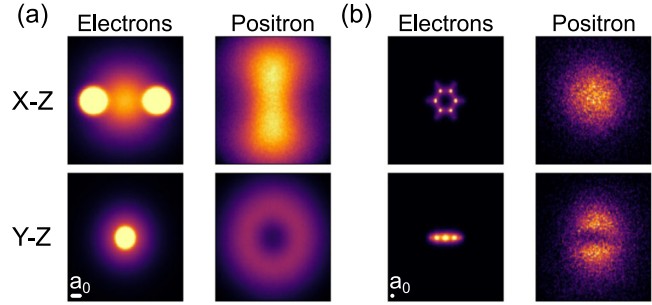

**Fig. 2 | Ground-state one-particle densities of positronic lithium and benzene.** Orthographic projections of the ground-state one-particle density for positronic (**a**) dilithium and (**b**) benzene molecules. Left/right columns show the electronic/positronic density. In the dilithium molecule, the positron is seen to be strongly localized to a torus wrapping the covalent bond. In the benzene molecule, the positronic component of the wavefunction is highly diffuse, resulting in a much noisier Monte Carlo estimate of the one-particle density. The positronic density resembles a $p$-orbital sandwiching the aromatic ring. The scale bar in the bottom left of each subfigure indicates the relative size of one Bohr radius, $a_0$.

**Table 2 | $2\gamma$ annihilation rates for positronic atoms and molecules accumulated with FermiNet wavefunctions, compared with those obtained via various other computational methods**

| Method | $2\gamma$ annihilation rate [ns$^{-1}$] | | | | | | |
|---|---|---|---|---|---|---|---|
| | HPs | [Na$^+$, Ps] | [Mg, e$^+$] | [LiH, e$^+$] | [BeO, e$^+$] | [Li$_2$, e$^+$] | [Benzene, e$^+$] |
| CI | 2.0183[26] | – | 1.000991[28] | 0.8947[a][31] | – | – | – |
| FN-DMC | 2.32[33] | – | – | 1.3602[a][34] | – | – | – |
| ECG-SVM | 2.4361[16], 2.4722[17], 2.4685[19] | 1.898[20] | 0.955[22], 1.0249[28] | 1.375[23] | – | – | – |
| Hofierka et al.[47] | – | – | – | 2.083 | – | – | 0.666 |
| FermiNet-VMC | 2.440(1) | 1.870(1) | 1.076(1) | 1.3391(7) | 0.9533(8) | 1.962(2) | 0.5247(6) |

[a]Contact density at equilibrium bond length of 3.348 bohr obtained by quadratic interpolation of three values around the minimum in the potential energy surface.
Statistical errors are omitted where they are smaller than the reported precision, or otherwise omitted in the referenced source. ECG results for all species besides HPs utilize the fixed-core approximation.

obtains more accurate total energies for the ground-state energy of the bare benzene molecule than FermiNet.

As with previous work utilizing FermiNet, computational scaling remains an issue. Calculations involving ≥50 particles are too expensive for presently available computational resources, prohibiting the application of the method to large molecules. One approach to calculating the ground-state wavefunctions of large molecules is to utilize pseudopotentials to remove core electrons from the calculations. The positronic component of the wavefunction is often small near the atomic nuclei, so we would expect the correlation effect between the core electrons and the positron to be small. Pseudopotentials have been successfully employed in previous QMC calculations of positron lifetimes in solids[44]. As the architecture employed herein only extends the original FermiNet architecture to treat different particle types separately, we expect that future advances in the computational efficiency of FermiNet and related neural network wavefunctions will be able to be combined with our approach to model positron binding.

These results demonstrate a general advantage of neural network wavefunction methods: the lack of a dependence on a basis set simplifies the treatment of systems which are not well described by traditional notions of atom-centered, low angular momentum basis functions. Highly accurate calculations can be carried out without foreknowledge of the appropriate features of the ground-state wavefunction, lowering the degree of inductive bias in the method. As a result, we believe that neural network variational Monte Carlo offers a highly performant, generic method for obtaining ground-state properties of any continuous-space Hamiltonian.

We have shown that the FermiNet ansatz for VMC calculations can be extended to include positrons naturally, treating positrons on an equal footing to electrons. Because this ansatz does not depend on a basis set, our treatment sidesteps traditional methods' issues in selecting and converging an appropriate basis set for describing positronic wavefunctions. Our method produces highly accurate results for several molecules with various binding mechanisms for positrons without any system-specific tuning. We expect that the simplicity of this method will lend itself to many challenging applications beyond those presented here, e.g., calculations involving multiple positrons. With additional computational effort, this method can provide accurate predictions for positron annihilation experiments.

## Methods

We find the ground-state wavefunction and corresponding ground-state energy of the many-body Coulomb Hamiltonian in the Born-Oppenheimer clamped nuclei approximation,

$$\mathcal{H} = -\frac{1}{2}\sum_i \nabla_i^2 + \sum_{i,j} \frac{q_i Z_j}{|\mathbf{r}_i - \mathbf{R}_j|} + \sum_{i>j} \frac{q_i q_j}{|\mathbf{r}_i - \mathbf{r}_j|} + \sum_{i>j} \frac{Z_i Z_j}{|\mathbf{R}_i - \mathbf{R}_j|}, \quad (1)$$

where $q_i, \mathbf{r}_i$ are the particle charges and positions, and $Z_i, \mathbf{R}_i$ are the charges and positions of fixed nuclei. Here, and throughout, we utilize Hartree atomic units: $\hbar = e = m_e = 1$. For the mixed electron–positron systems considered here, $q_i = \pm 1$. We solve for the many-body ground-state wavefunction using the variational Monte Carlo (VMC) method[56]: a many-body wavefunction $\Psi_\theta$, parameterized by $\theta$, is continuously updated via a gradient descent procedure to minimize the energy expectation value,

$$\langle E \rangle_\theta = \frac{\int \Psi_\theta^*(\mathbf{r}) \mathcal{H} \Psi_\theta(\mathbf{r}) d\mathbf{r}}{\int \Psi_\theta^*(\mathbf{r}) \Psi_\theta(\mathbf{r}) d\mathbf{r}}, \quad (2)$$

where $\mathbf{r} = (\mathbf{r}_1, \ldots, \mathbf{r}_N)$. This integral, and its gradient with respect to $\theta$, are evaluated via Monte Carlo integration. Particle configurations, $\mathbf{r}^{(i)}$, are sampled from the probability density $|\Psi_\theta(\mathbf{r})|^2$ via the Metropolis-Hastings algorithm. The expectation value of the energy and its gradient are then evaluated by the Monte Carlo estimators,

$$\langle E \rangle_\theta = \lim_{N\to\infty}\left[\frac{1}{N}\sum_{i=1}^N \frac{\mathcal{H}\Psi_\theta(\mathbf{r}^{(i)})}{\Psi_\theta(\mathbf{r}^{(i)})}\right] = E_{\mathbf{r}\sim|\Psi_\theta|^2}\left[\frac{\mathcal{H}\Psi_\theta(\mathbf{r})}{\Psi_\theta(\mathbf{r})}\right] \quad (3)$$

$$\nabla_\theta \langle E \rangle_\theta = 2E_{\mathbf{r}\sim|\Psi_\theta|^2}\left[\left(\frac{\mathcal{H}\Psi_\theta(\mathbf{r})}{\Psi_\theta(\mathbf{r})} - \langle E \rangle_\theta\right)\nabla_\theta \log|\Psi_\theta(\mathbf{r})|\right] \quad (4)$$

The FermiNet represents the many-body wavefunction as a sum of block-diagonal determinants containing many-particle functions which depend upon the coordinates of all particles in a permutation equivariant manner. This is written

$$\Psi(\mathbf{r}) = \sum_k^{n_{\det}} \prod_\chi \det\left[\psi_i^{k\chi}\left(\mathbf{r}_j^\chi; \left\{\mathbf{r}_{/j}^\chi\right\}; \left\{\mathbf{r}_j^{/\chi}\right\}\right)\right], \quad (5)$$

where the set $\{\mathbf{r}_{/j}\}$ includes all particle coordinates except $\mathbf{r}_j$, and $\chi = (\sigma, q)$ labels species of particles which are distinguished by their spin $\sigma \in (\uparrow, \downarrow)$ and charge $q \in (+, -)$. Here we have made a slight abuse of notation for the sake of brevity: permutation invariance for the set $\{\mathbf{r}_j^{/\chi}\}$ is only maintained between particles of the same species. We emphasize that these are not the dense determinants discussed in recent works extending FermiNet[57], except the benzene calculation for which dense determinants were used for the electronic component of the wavefunction. In this case, the determinant is not block-diagonal between the two electronic spin species. The many-particle functions $\psi_i$ are represented by a deep neural network[48] (architecture described in the Supplementary Material). Multiplicative coefficients are omitted from the sum as they are trivially absorbed into the entries of the determinant. Gradient descent is performed via the Kronecker-factored approximate curvature (KFAC) algorithm, an approximation of natural gradient descent[58] which scales well to large neural networks[59]. Natural gradient descent is closely related to the stochastic reconfiguration method, well-known in the quantum Monte Carlo literature[60]. The present work introduces two alterations to the original FermiNet architecture. Firstly, we have included positronic functions as additional species, i.e. additional blocks in the determinant. Secondly, we utilize distinct weights in the neural network layers for every species (unlike the original FermiNet, where spin up and down electronic orbitals shared weights).

A single determinant of the form in Eq. (5) can represent any fermionic many-body wavefunction[61]. In practice, the argument for the universality of FermiNet determinants depends upon the representation of discontinuous functions which cannot be constructed using realistic neural networks. Despite this, FermiNet-VMC calculations obtain state-of-the-art accuracy in ground state energy calculations for a range of molecules and solids[48,49,57,62–64] with a linear combination of a small number of determinants. We utilize 32 determinants for all calculations in the present work.

We only consider calculations involving a single positron in the present work. We have discussed the treatment of the positronic spin coordinate only to demonstrate how our technique may be extended to calculations involving many positrons, as such systems have recently attracted theoretical interest.

Ground-state wavefunctions for bare and positronic molecules are not guaranteed to be similarly converged after an equal number of gradient descent steps. This introduces uncontrolled error in calculating positron binding energies via VMC. Previous work has shown that FermiNet-VMC calculations yield ground-state energies within chemical accuracy (~1.5 milliHartrees) of exact results for many small molecules[48,49]. With this level of accuracy, the uncontrolled error will be negligible for molecules with a large positron binding energy. However, for molecules with very small binding energies, or large molecules for which the uncontrolled error may become large

compared to the positron binding energy, there is no guarantee that an accurate estimate of the positron binding energy will be obtained by comparing ground-state calculations of different quality. In these cases, we employ the variance matching technique described by Entwistle et al.[65], addressed in the Supplementary Material.

## Data availability

Source data has been deposited in Figshare under accession code https://doi.org/10.6084/m9.figshare.25109330[66]. Ground-state energies used to produce other results (e.g. equilibrium bond lengths) which are not directly quoted in the main text are provided in the Supplementary Material.

## Code availability

The results presented in this study were obtained using a private fork of the public FermiNet repository[67], available under the Apache-2.0 license. The modifications to the code required to reproduce the results in this paper are being prepared for release in the public repository.

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

## Acknowledgements

This work was undertaken with funding from the UK Engineering and Physical Sciences Research Council (EP/T51780X/1) (GC). Calculations were carried out with resources provided by the Baskerville Accelerated Compute Facility through a UK Research and Innovation Access to HPC grant. Additionally, the authors gratefully acknowledge both PRACE, and the Gauss Centre for Supercomputing e.V. (www.gauss-centre.eu) for funding this project by providing computing time through the John von Neumann Institute for Computing (NIC) on the GCS Supercomputer JUWELS at Jülich Supercomputing Centre (JSC). Via his membership of the UK's HEC Materials Chemistry Consortium, which is funded by EPSRC (EP/R029431), WMCF used the UK Materials and Molecular Modelling Hub for computational resources, MMM Hub, which is partially funded by EPSRC (EP/T022213).

## Author contributions

All of the experiments and the results presented in this work were performed by G.C., with the academic supervision of W.M.C.F., D.P., and J.S.S.

## Competing interests

The authors declare no competing interests.
