## [Peer Review File · Nature Communications]

Neural network variational Monte Carlo for positronic chemistryReviewer #1 (Remarks to the Author):

This work presents AI-based calculations for positronic molecules. The implemented method based on the variational Monte Carlo (VMC) and Neural network (FermiNet) is a versatile and accurate quantum mechanical calculation, originally proposed for ordinary atomic/molecules. The authors extend this approach to the positronic molecules which include an anti-particle of the electron. It has been known that the positronic wavefunction has a diffuse structure keeping a high correlation with electrons which makes the ordinary quantum chemical calculations difficult to achieve accuracy. The presented calculations for PSH, LiH, BeO, Li dimer, and benzene predict the positron affinity with good accuracy. The results will contribute to exploring positron chemistry in cooperation with recent experiments.

I would like the authors to consider the following points:

- 1) The first paragraph of the Introduction describes the recent progress in experimental studies on enhanced positron annihilation with isolated molecules. The enhanced annihilation with the molecules is explained by a vibrational Feshbach resonance where the positron attachment to the molecule is coupled with the vibrational excitations of the target molecule. In the second paragraph, the authors mentioned the importance of enhanced annihilation in applications, e.g. PET, defect analysis, and astrophysics. However, the PET at the current stage does not explicitly utilize the "enhanced" annihilation. In addition, the spatial resolution of the PET is still far from the atomic length scale. Similarly, the defect analysis of materials utilizes the annihilation lifetimes and gamma-ray energies (or angular correlations) which are not directly related to the Feshbach resonances. The reference of astrophysics, the observation of Ps annihilation in the galaxy, does not assume the positron annihilation with molecules. I agree with the promising future in exotic antimatter chemistry and the important role of ab initio calculations, but the connection between the first and second paragraphs seems to be somewhat misleading.
- 2) For the readability of Figure 1 (a), it would be worth adding the nuclear positions in the figure because the interatomic distances are varied.
- 3) On page 4, the authors state that the positron affinity of benzene is obtained within chemical accuracy, but the accuracy is insufficient for other species of experimental interest. It would be better to clarify what the "other species" indicate.
- 4) It might be worth commenting on the prospects for annihilation rate calculation because one of the motivations of this study comes from the positron annihilation with molecules.

Reviewer #2 (Remarks to the Author):

All comments are in the uploaded file (review.txt).

Reviewer #3 (Remarks to the Author):

This paper is about calculating positronic wave functions with neural network wave functions. The work here is less of a breakthrough than previous FermiNet papers and more of a technical paper that is likely to only appeal to experts. The paper is written as a response to some extent to Hofierka et. al. which is reference 36 in the manuscript. Many results are compared between the two papers. For one of the main results, LiH, the authors compare to a result from 2001 (ref 18) with the ECG method, in which they are not even fully able to surpass the results from over 20 years ago. Finally, the analysis and discussion in this manuscript is presented with limited detail.

There aren't many noteworthy results. Most of the reported calculations in the results section are benchmarking against previous methods. The paper is written as a technical benchmarking paper. While I think FermiNet might be useful for positronic chemistry, it is not enough for the authors to claim a breakthrough just because they used modern techniques such as neural networks. As

mentioned, not only in the case of LiH are the authors results comparable to results and methods from over 20 years ago, but the cost of the neural network approach is also not well explored, at least in comparison to other techniques.

The authors also say they aren't using state of the art neural network methods. Why not use psiformer or other more advanced neural network wavefunctions for these calculations? Also, it is not clear that the authors provide any access to these codes for positronic wave functions, and it is likely that the techniques presented here are out of reach for most researchers in the field.

Regardless, the paper does demonstrate an interesting set of technical result for experts in the field. However, there does not appear to be any new physical results in the paper. The paper does not use the best techniques available for the neural network wavefunctions, such as psiformer. Overall, I would suggest this be published in a more specialized journal.

Finally, the authors could proofread the paper more. I didn't see too many typos, but there are a few places which can be improved upon such as on page 4 where the following was written:
"computaitonal chemistry methodsis a signi."

We thank all of the referees for their insightful and constructive comments, which have led us to carry out a number of additional calculations and refine the manuscript substantially. Your expert opinions have significantly improved the quality of our work, and we look forward to receiving further feedback.

Reviewer #1:

We would like to offer specific thanks to reviewer #1 for prompting us to carry out additional calculations of positron annihilation rates. We were delighted to see that these results compare favorably to previous calculations. Additionally, your comments have helped us to substantially clarify our introduction. We have addressed your report point-by-point below.

This work presents AI-based calculations for positronic molecules. The implemented method based on the variational Monte Carlo (VMC) and Neural network (FermiNet) is a versatile and accurate quantum mechanical calculation, originally proposed for ordinary atomic/molecules. The authors extend this approach to the positronic molecules which include an anti-particle of the electron. It has been known that the positronic wavefunction has a diffuse structure keeping a high correlation with electrons which makes the ordinary quantum chemical calculations difficult to achieve accuracy. The presented calculations for PsH, LiH, BeO, Li dimer, and benzene predict the positron affinity with good accuracy. The results will contribute to exploring positron chemistry in cooperation with recent experiments.

I would like the authors to consider the following points:

1) The first paragraph of the Introduction describes the recent progress in experimental studies on enhanced positron annihilation with isolated molecules. The enhanced annihilation with the molecules is explained by a vibrational Feshbach resonance where the positron attachment to the molecule is coupled with the vibrational excitations of the target molecule. In the second paragraph, the authors mentioned the importance of enhanced annihilation in applications, e.g. PET, defect analysis, and astrophysics. However, the PET at the current stage does not explicitly utilize the "enhanced" annihilation. In addition, the spatial resolution of the PET is still far from the atomic length scale. Similarly, the defect analysis of materials utilizes the annihilation lifetimes and gamma-ray energies (or angular correlations) which are not directly related to the Feshbach resonances. The reference of astrophysics, the observation of Ps annihilation in the galaxy, does not assume the positron annihilation with molecules. I agree with the promising future in exotic antimatter chemistry and the important role of ab initio calculations, but the connection between the first and second paragraphs seems to be somewhat misleading.

We agree that the wording of the opening paragraphs unintentionally implies a link between the enhanced annihilation rates for positrons forming bound states with molecules and the technological applications of positrons we highlighted. We have rearranged and reworded these paragraphs to make it clear that these are unrelated, and that our work is only concerned with the specific problem of providing predictions for positron attachment to molecules. The reference to these disparate fields is intended only to emphasize to a general audience that positrons are more than just a curiosity for theoretical physicists.

2) For the readability of Figure 1 (a), it would be worth adding the nuclear positions in the figure because the interatomic distances are varied.

We have added markers to indicate the fixed positions of the nuclei relative to the scaled axes of each panel of the figure and adjusted the caption accordingly.

3) On page 4, the authors state that the positron affinity of benzene is obtained within chemical accuracy, but the accuracy is insufficient for other species of experimental interest. It would be better to clarify what the "other species" indicate.

As a concrete example, we have highlighted a reference to Young and Surko (2008), where at least 10 small molecules are found to possess a positron binding energy below the threshold for chemical accuracy:

“Many chemical species possess positron binding energies below chemical accuracy (see, e.g., the experimental binding energies in ref. [11], where 15 small organic molecules are found to possess binding energies below chemical accuracy).”

4) It might be worth commenting on the prospects for annihilation rate calculation because one of the motivations of this study comes from the positron annihilation with molecules.

Although we originally chose to focus on benchmarking ground-state energy calculations to enable a variational comparison between methods, annihilation rates can be obtained within the variational Monte Carlo framework. Since receiving your report, we have used the trained wavefunctions for every molecule studied to calculate annihilation rates and added these to the manuscript, including a comparison to annihilation rates in the existing literature, in the new Table II and surrounding text. We describe the procedure for calculating the annihilation rate in a new section in the Supplementary Material.

Reviewer #2:

We would like to thank reviewer #2 for offering a keen critical eye to many areas of the manuscript, but particularly in regards to our discussion of the challenges for traditional methods (specifically, the configuration interaction method). Your comments have deepened our understanding of the history of quantum chemical calculations of positron binding, and given us a greater appreciation of the advantages our *own* method can offer. We have addressed your report point-by-point below.

Proper description of the positron binding to normal matter is a challenging task for computational methods of quantum chemistry. Since a breakthrough that appeared a quarter of century ago, and resulted from applications of explicitly correlated Gaussian basis functions in the variational framework, and Diffusion Quantum Monte Carlo simulations, for few-electron positronic systems, further progress is rather slow. In the work under review, the calculations are reported for a number of positronic bound states, carried out within the variational Monte Carlo framework, with the wavefunctions generated by neural network algorithm. Systems with various properties were studied, including positronic atoms (PsH, e^+Na , e^+Mg), polar molecules (e^+LiH , e^+BeO), and molecules without permanent dipole moment (e^+Li_2 , $e^+C_6H_6$). Theoretical results for the latter class of systems were not published previously, and these for positronic benzene may be compared, and are in reasonable agreement, with experimental Measurement.

Unfortunately the article contains statements that are either unclear or even misleading. The results in the field of reliable prediction of the positron binding to large, organic molecules, might however appear very important. Therefore an opportunity should be given to the Authors, to revise the text, clarify and rectify the elements that raise doubts, and finally send it to re-evaluation by referee(s).

Let me begin with the general layout, which looks quite exotic, at least for a person who has read hundreds of scientific papers. Results and discussion following immediately the introduction resulted in initial impression that I read a sponsored article! In my opinion, the section in which the methods used in the research are presented, should not be postponed to the end of the text, but precede the results. On the other hand, if the Editor agrees for such unusual order of sections, I will not oppose as well. There are other points, that are to be addressed by the Authors.

Although we sympathize with the referee's opinion on the layout of our article, we are using the layout requested for manuscripts submitted to Nature Communications. From the Nature Communications submission guidelines (<https://www.nature.com/ncomms/submit/article>):

"The main text of an Article should begin with a section headed Introduction of referenced text that expands on the background of the work (some overlap with the abstract is acceptable), followed by sections headed Results, Discussion (if appropriate) and Methods (if appropriate)."

1. Introduction

I would not be sure whether referencing medical physics (as an area of applications of positron annihilation) papers does make a sense in the context of the work devoted to bound states of the positron with "normal" matter. Physicians are not interested in these states. They want to know to which part of the body the radioactive isotope they inject is transported. Similarly, one could cite electrotechnics journals, because electricians deal with the electrons too.

We chose to highlight some broader technological applications of positrons to emphasize to a general audience that positrons are more than a curiosity for theoretical physicists (we note that the excellent thesis of M.W.J. Bromley, which you recommended to us later in your report, includes similar expository mentions of these fields in the introduction). On the suggestion of Referee 1, we have rewritten the opening paragraphs of the introduction to make it clear that the present work is concerned only with the description of the bound states of positrons and ordinary molecules. We have, however, chosen to retain a brief reference to these other fields: when discussing this work with our colleagues we are often met with surprise that positrons are technologically relevant at all!

The need for diffuse basis functions, and centring these function outside of the nuclei, have nothing to do with slow convergence of CI calculations for positronic systems. The real reason is the difficulty of the description of the electron-positron cusp in the wavefunction. The basis functions with large angular momenta are needed for reasonably good mirroring of the partial wave expansion of the wavefunction, and not due to the repulsive potential of the nuclei. I would advise looking into the PhD thesis of M.W.J. Bromley (Northern Territory University, Australia), probably still to be found in a public repository. Obtaining the bound state of $e+Li$ (S symmetry, effective $Li+$ core potential, so only 2 "active" particles) with sufficiently low energy (below -0.25 hartree) required the orbitals with $l=29$!

We have amended our explanation of the slow convergence of CI calculations with basis set size to clarify that accurately describing electron-positron correlations is more relevant than the repulsion between the nuclei and the positron (paragraph 5 introduction) .

Regarding the inclusion of diffuse basis functions, we refer mainly to the work of RJ Buenker, e.g., *Buenker and Liebermann, Nucl. Instr. and Meth. in Phys. Res. B 266 (2008) 483–490*):

"A large number of diffuse functions are added, however, that are needed for an accurate representation of the lone positron orbital in each case."

Similar discussion regarding the augmentation of the basis with additional diffuse functions centered on the most electronegative atom (these works are concerned with strongly dipolar molecules where this is the binding center of the positron) can be found in, e.g., *Buenker et al. J. Phys. Chem. A 109, 5956 (2005)*, *Buenker et al. J. Chem. Phys. 126, 104305 (2007)*.

We have reworded the fifth paragraph of the introduction to make it clear that the slow convergence of CI calculations with the basis is due to the requirement for basis functions with large angular momenta, but we retain the mention of the diffuse nature of the positronic density posing additional difficulties for methods relying on a set of basis functions. We have also added citations to the very helpful work of Bromley and collaborators, and thank you for bringing them to our attention.

2. Methods (this is the order of the material I prefer)

Many-electron, or many-particle orbitals? This notion is self-contradictory! Orbital always means a function of spatial coordinates of only one particle, and there is no place for "licentia poetica" in scientific papers. I felt confused while reading the paragraph on the representation of the wavefunction in FermiNet, and then realized that in the original article, published in Physical Review Research (Ref. 37), many-electron functions are mentioned, which are the correct term for functions dependent on the coordinates of more than two electrons (geminals may be used for two particles). "Many-electron orbitals" appear in the preprint (Ref. 47) which apparently has not been published (yet). Please, use many-particle functions instead.

We have rephrased all of the mentions of "many-x orbitals" to "many-x functions".

How is the wavefunction (or basis functions) represented, for the integration? Does the neural network algorithm produce the values of the basis functions in a set of points ("samples") for further calculation of the integrals, without referencing an analytical formula? The integrals would be then calculated with a separate code. Such a procedure

would look logical for VMC, but it is explained neither in the main text nor in the appendix, so maybe I'm wrong. Please answer this question.

Our wavefunction ansatz does not reference a set of basis functions. The many-particle functions appearing in the determinants are the outputs of a neural network. Phrased differently, the neural network is a parameterized mapping from the particle coordinates to the values of a set of permutation-equivariant functions which are used to construct determinants, producing a totally antisymmetric result. The determinants are not Slater determinants because the entries are not single-particle orbitals. As shown in our first paper on neural wavefunctions (Ref. 48), a single many-body determinant of this type can in principle represent any antisymmetric function, although in practice the accuracy is improved by using a small number of determinants. One can think of the neural network, including the determinant evaluation, as a black box that returns the value of $\Psi(r_1, r_2, \dots, r_N)$ for any set of particle positions $\{r_1, r_2, \dots, r_N\}$ provided as input. The network is analogous to, but much more general than, the trial ground-state wavefunctions used in traditional variational / diffusion Monte Carlo calculations.

As in traditional variational Monte Carlo, the high dimensional integral required to evaluate the expectation value of the energy is calculated by direct Monte Carlo integration. Electron (plus, in this case, positron) positions (r_1, r_2, \dots, r_N) are sampled from $|\Psi|^2$ via the Metropolis-Hastings algorithm. No fixed or pre-chosen grid of points is used. Likewise, gradients of the expectation value of the energy can be formulated as Monte Carlo estimators, which are accumulated in the same way as the energy. The expectation value of the energy is minimized by applying a gradient descent procedure to the parameters of the neural network using the Monte Carlo estimates of the gradient. This procedure is quite different from the evaluation of the expectation value of operators in a CI framework: the high dimensional integrals are not broken down in terms of integrals of constituent basis functions.

We have expanded our description of the VMC method in the Methods section to clarify how the energy and its derivatives are calculated within this framework, including the mathematical details of the algorithm we have sketched out in the preceding paragraphs.

Going further, how many determinants were generated (n_{det} in equation 3)? Only one, according to the paragraph beginning with "A single determinant of the form in Eq. (3) can represent any fermionic many-body wavefunction", or a larger number?

We utilized an expansion of 32 determinants for every calculation presented in the paper. We apologize for the omission of this important detail from the paper and have inserted it into both the main text and the list of hyperparameters in the Supplementary Material.

Concerning the appendix, how to read equations (4), and (5)? I see such notation ("vector,euclidean norm of this vector") for the first time. Maybe there are holes in my education, but it is better to admit lack of knowledge than to try to look wise. I might only guess the meaning of this notation (extra elements of the vector, appended after the coordinates?) and other readers could have similar problem. Please give at least the reference to proper handbook. Please define r_l vectors as well.

This notation has become a common shorthand in papers describing neural network variational Monte Carlo, so we forget that it is quite non-standard and easily confusing.

We have adjusted the text around these equations to clarify the points that you have raised.

At the end of the appendix, there is (possibly) alarming information that single precision FP numbers were used in all calculations. ALL. I know that neural networks usually work in low precision, but such precision may be insufficient for quantum-chemical calculations. Single precision numbers have 24 mantissa bits, which translate to 7, or at most 8 decimal digits. Were the Monte Carlo integrations also carried out in single precision? If so, then what about the rounding errors, whose accumulation leads to further loss of precision? The binding energy of positronic benzene is obtained at 6'th significant digit (total energy: c.a. 232 hartrees, binding energy: 4 millihartrees). I need convincing arguments (literature reference?) that this result is something more than a trained accumulation of rounding errors. Or maybe the integrations were performed in double precision?

We have in the past tested the effect of running calculations in double precision and found that the changes are very small on the scale of accuracy required for this work. For these reasons, we do not think that our results are substantially influenced by floating point error.

In response to your query, however, we re-accumulated the positronic ground-state energies of the beryllium oxide (at equilibrium separation) and benzene molecules in both single and double precision, obtaining the following results:

Molecule	Floating point precision	Ground-state energy [Ha]
[Benzene, e+]	Single	-232.2230(3)
[Benzene, e+]	Double	-232.2227(3)
[BeO, e+]	Single	-89.93076(6)
[BeO, e+]	Double	-89.93081(6)

The obtained energies are equal to within statistical error bars (we did not use a fixed seed for random number generation, and GPU calculations are well-known to exhibit non-determinism due to the highly parallel execution model; hence the variation in the final Monte Carlo estimates). This indicates that, to the accuracy required for the reported binding energies, our results are not influenced by the accumulation of floating point errors. Testing the inference energies in this way does not discount the possibility that the *training* is biased by floating-point errors, but it guarantees that all of the final results presented in the paper are accurate.

3. Results

The results look promising, but are they reliable? It is to be appreciated that the energies of polar, positronic molecules, known from other studies, are well reproduced. Answer the questions related to the methods and appendix first. Not only in the response letters, but in the article text too.

We are not sure how to respond to your question except with a resounding yes! If you are asking whether our VMC-based wavefunction optimization method produces reproducible results then the answer is again yes. If we reoptimize from scratch, starting with a different random number seed, we converge to very slightly different results, of course, but the differences are so small as to be negligible.

We hope that our answers to your questions, and the adjustments to the manuscript made at the recommendation of all of the referees, are satisfactory. In particular, we would like to highlight that we have expanded the results presented in the manuscript to include calculations of the positron annihilation rate for all of the species studied. The annihilation rates we calculate compare favorably with ECG results for the small systems (positronium hydride, lithium hydride, and the alkali metal atoms) where ECG calculations are viable; this suggests that our method describes the coalescence between electrons and positrons quite accurately.

Reviewer #3:

We would like to thank reviewer #3 for prompting us to refine the message of the paper. As a result of your comments, we have modified the text to clarify that these results offer important insights into neural network wavefunction-based methods. We have addressed your report point-by-point below.

This paper is about calculating positronic wave functions with neural network wave functions. The work here is less of a breakthrough than previous Ferminet papers and

more of a technical paper that is likely to only appeal to experts. The paper is written as a response to some extent to Hofierka et al. which is reference 36 in the manuscript.

The comment that our manuscript is written to some extent as a response to Hofierka *et al.* is not unwarranted. We saw the statement that QMC methods have not to date been particularly successful in describing bound states between positrons and ordinary matter as an interesting challenge for our recently developed neural network quantum Monte Carlo methods.

Not only are we able to demonstrate that neural network quantum Monte Carlo is capable of obtaining extremely accurate results for the ground-state energy and positron binding energy for a large number of molecules, but this problem domain provides valuable insight into the reason why these methods are so powerful compared to traditional variational Monte Carlo methods. The lack of a basis set dependence for neural network wavefunctions immediately resolves significant difficulties faced by traditional wavefunction ansätze in obtaining appropriate orbitals for populating the Slater determinants used in the majority of conventional quantum chemistry methods. We believe that this is of significant interest to a more general audience, in addition to our excellent quantitative results that will be of interest to researchers in the field of positronic chemistry. Furthermore, our method is in principle straightforward to extend to the solid state, where the interaction between positrons and defects in semiconductors is of great technological interest.

More broadly, our results demonstrate that neural network quantum Monte Carlo is an effective tool for obtaining the ground-state properties of systems beyond purely electronic Hamiltonians. At the time of writing, it is not clear that there is a regime where the performance-cost tradeoff of neural network quantum Monte Carlo is superior to any other method for typical molecules and solids. However, there are many problem domains where, as in the case of positron-matter interactions, traditional methods fall short. Thus, the demonstration that neural network quantum Monte Carlo may yield accurate ground-state properties of non-electronic Hamiltonians dramatically broadens the scope of this approach.

We have modified the abstract and discussion of the manuscript to place greater emphasis on these aspects of our work.

Many results are compared between the two papers. For one of the main results, LiH, the authors compare to a result from 2001 (ref 18) with the ECG method, in which they are not even fully able to surpass the results from over 20 years ago.

We compare only the binding energies for lithium hydride and benzene with the results from Hofierka *et al.* The other species we have studied are compared with ground-state energies and binding energies obtained with configuration interaction methods, quantum Monte Carlo methods, and the (fixed-core) stochastic variational method with explicitly correlated Gaussians.

Our calculation of the ground-state energy of positronic lithium hydride offers a variational improvement over the result of Strasburger (ref. 18) by 0.3 milliHartrees, and our calculation of the ground-state energy of the bare molecule agrees with the ECG result (for the largest basis employed) to within 0.03 milliHartrees. Thus, Strasburger's calculation of the positron binding energy is considerably less accurate than ours.

The ECG result for the ground state of the bare LiH molecule is within ~ 0.01 milliHartrees of the energy extrapolated to the complete basis set limit (note that such an extrapolation is non-variational), suggesting that the ECG method is remarkably accurate in this case. There is almost certainly very little improvement to be gained. As our approach is stochastic, with the network optimization and energy evaluations subject to random noise, it is difficult to achieve the precision that would be required to challenge ECG in this example. Nevertheless, the improvement in the energy of positronic lithium hydride provides strong support for the value of our approach in more complex systems.

Finally, we stress that the ECG method can only be applied to tiny systems. Calculations for LiH are possible because of the small number of electrons. ECG calculations can be carried out for larger atoms such as Mg using the fixed-core approximation to reduce the number of valence electrons to 2, but cannot be used to calculate the positron binding energy of any but the simplest molecules. For instance, the ECG method will never be used to calculate the positron binding energy of benzene. We include comparisons to ECG results in our work as they offer highly accurate benchmarks for the smallest systems.

Finally, the analysis and discussion in this manuscript is presented with limited detail.

In response to specific areas highlighted for improvement by the referees, we have expanded several parts of the text. The most substantial change is the inclusion of calculations of positron annihilation rates.

We find that our method produces positron annihilation rates in good agreement with ECG results (which should be the most accurate for this non-variational quantity, given that ECG wave functions describe the coalescence of the positron and electron quite well) for HPs and LiH. Interestingly, the method of Hofierka *et al.* overestimates the positron annihilation

rate of LiH compared to the ECG benchmark, suggesting that our approach may be better suited for the calculation of this quantity.

Additionally, we have expanded the discussion around the implications of our results on the broader applicability of neural network variational Monte Carlo to non-standard Hamiltonians.

There aren't many noteworthy results. Most of the reported calculations in the results section are benchmarking against previous methods. The paper is written as a technical benchmarking paper. While I think Ferminet might be useful for positronic chemistry, it is not enough for the authors to claim a breakthrough just because they used modern techniques such as neural networks. As mentioned, not only in the case of LiH are the authors results comparable to results and methods from over 20 years ago, but the cost of the neural network approach is also not well explored, at least in comparison to other techniques.

This work constitutes a vast improvement in the range of positronic systems and the predictive accuracy with which they can be described using quantum Monte Carlo methods. Furthermore, we would go so far as to claim that this is amongst the first works to demonstrate a genuine advantage of approaches based on neural wave functions over traditional quantum chemical approaches. The ability to optimize neural wave functions without having to choose a basis set is particularly advantageous when, as in positronic systems, the basic chemistry is not very well understood. As mentioned, we have adjusted the text of the manuscript to place greater emphasis on this fact, which we believe to be of interest to a general audience.

As previously discussed, the LiH results from Ref 18 are of extremely high quality; with only four electrons it is possible to obtain almost exact (and hence almost unimprovable) results using an explicitly correlated Gaussian basis. It should be noted, however, that our results for the slightly more complicated $[\text{LiH}, e^+]$ complex are considerably more accurate than those of Ref 18, and that our method is capable of scaling far beyond the scope of ECG calculations.

The cost of neural network quantum Monte Carlo has been explored previously. In the original FermiNet paper, we demonstrated the expected $O(N^4)$ scaling of the time per training iteration with electron number N . However, we are not yet operating in the regime where the asymptotic scaling is limiting; rather the computational cost of these calculations is dominated by a large prefactor in the scaling. Since the original FermiNet paper was released, substantial reductions in the computational cost of neural network quantum Monte Carlo have been achieved by various authors, and we expect that this cost will only continue to be reduced in the coming years.

The authors also say they aren't using state of the art neural network methods. Why not use psiformer or other more advanced neural network wavefunctions for these calculations? Also, it is not clear that the authors provide any access to these codes for positronic wave functions, and it is likely that the techniques presented here are out of reach for most researchers in the field.

At the time the calculations were performed, FermiNet was the state-of-the-art neural network quantum Monte Carlo method. The technical implementation of our modifications to the FermiNet, and the earliest calculations for positronic systems, were carried out in the summer of 2022. The PsiFormer paper was not released publicly until November 2022, and the open-source implementation of the PsiFormer was released in March 2023, by which point the results obtained using the FermiNet had been finalized and we were preparing the manuscript.

We have a strong track record of open-sourcing our code. We have released open-sourced codes for reproducing the results in the original FermiNet paper, the PsiFormer paper, and our work on the electron gas (Cassella 2023). We will release an open-source implementation of our modifications to the FermiNet for application to positronic systems when this manuscript is accepted for publication.

We are in the process of implementing positronic calculations with the PsiFormer. The substantial differences between the FermiNet and PsiFormer architectures mean that it is not straightforward to copy our modifications from FermiNet. Crucially, the PsiFormer lacks FermiNet's 'two-electron stream' and thus requires the inclusion of an explicit Jastrow factor to represent the cusps in the wavefunction at the coalescences of pairs of particles. Getting this right is extremely important for describing positronic wave functions accurately, suggesting that a FermiNet-style ansatz with an explicit two-particle stream is in some ways *more* suitable for this problem. So far, we have been able to use the PsiFormer to reproduce our FermiNet results for positronium hydride and dilithium, but the FermiNet results are still (variationally) better for the other positronic systems studied in the present manuscript. Obtaining a robust ansatz for positronic calculations based on the PsiFormer will be the subject of future work.

While neural network quantum Monte Carlo does require access to GPU compute, we do not believe that the resources required to reproduce our results or study similar systems are beyond the reach of other researchers in the field. The computational resources used to carry out our calculations were awarded via the publicly accessible EuroHPC grant system, open to any European academic. All of the calculations except those for benzene can be obtained in less than a week using a single GPU costing a few thousand euros. Since this work was carried out, the computational cost of such calculations has been approximately

halved due to advances in the algorithmic evaluation of neural network laplacians, available through the open-source FOLX library from Microsoft (<https://github.com/microsoft/folx>).

Regardless, the paper does demonstrate an interesting set of technical result for experts in the field. However, there does not appear to be any new physical results in the paper. The paper does not use the best techniques available for the neural network wavefunctions, such as psiformer. Overall, I would suggest this be published in a more specialized journal.

Neural network variational Monte Carlo is a rapidly growing field. At the time of writing, the original FermiNet paper has accumulated several hundred citations, and our work on the electron gas has been cited 41 times (according to Google Scholar) since its publication last year. There is a broad general interest in these techniques and their advantages. As we have mentioned, this work builds upon our understanding of the advantages of neural network quantum Monte Carlo in a sense that will be of interest to this general audience.

Finally, the authors could proofread the paper more. I didn't see too many typos, but there are a few places which can be improved upon such as on page 4 where the following was written:

"computaitonal chemistry methodsis a signi."

We have corrected this typo in the revised manuscript.

In response to comments from other referees, we have expanded our results to include calculations of the positron annihilation rates for all of the species studied. This quantity is extremely sensitive to the quality of the wavefunction, and our results compare very favorably with the reliable ECG results for positronium, lithium hydride, and the alkali metal atoms.

Additional changes

While revising the manuscript, we made additional corrections, mainly regarding the reporting of benchmark values in the Results section. None of these changes alter the conclusions of the original work:

- Citations have been inserted for the positron affinity of the sodium and magnesium atoms in the text and for the positron affinity of the beryllium atom in the caption of Figure 1.
- The result of a calculation using model potentials for the positron affinity of the sodium atom, which was incorrectly quoted as a Cl result, has been removed.

- The SJ-FN-DMC ground-state energy for LiH was reproduced incorrectly from Kita *et al.* (listed in the same table in the original work, from an earlier calculation). This has been updated to the correct value, and the corresponding binding energy has been updated accordingly.

Reviewer #1 (Remarks to the Author):

The authors have revised and improved their manuscript with a more clear description of their objectives, motivations, methods, and new results on annihilation rates. They have shown a detailed explanation for the reviewers' comments. In particular, the accuracy of the annihilation rate which is added in this revision well demonstrates their excellent description of attractive behavior between the electron and positron because the annihilation rate requires higher accuracy to be converged than the energy itself. Notably, the FermiNet-VMC provides the annihilation rate of HPs with higher accuracy (closer to ECG-SVM) than the Fixed Node DMC calculation. The code availability in the public repository will also accelerate the research in positronic chemistry. The authors' replies are considered satisfactory to the reviewers' comments. I would like to comment on the sentence in the Introduction: "The formation of these bound states, enabled by an incident positron exciting a vibrational Feshbach resonance, results in greatly enhanced annihilation rates." Strictly speaking, the vibrational Feshbach resonance states are not bound states that the authors have exactly calculated. The vibrational Feshbach resonance state is surely a quasi-bound state but is decaying into the lower vibrational state + positron.

Reviewer #2 (Remarks to the Author):

Review of revised article "Neural network variational Monte Carlo for positronic chemistry", by Gino Cassella, W. M. C. Foulkes, David Pfau, and James S. Spencer.

The article, in which the calculations for positronic systems are presented, including non-polar molecules (Li₂, C₆H₆), has been modified accordingly to reviewers' remarks. I accept the arguments of the Authors; in my opinion, most problematic elements of the text are clarified, and the changes head in the right direction. Concerning the non-orthodox presentation order of the scientific material - if this is the Editor's requirement, then I have nothing to say. This paper could be accepted for publication, but after further revision - this time a minor one. My present remarks, to be considered by the Authors, are as follows:

1. I request absolutely the table with single and double precision results, as provided in the rebuttal letter, to be included in the article, with proper comment. I find this information crucial and so could many other readers, that the calculations involving millions (billions?) of summations, carried out on the edge of available precision, are sufficiently numerically stable, and their results do not sink in accumulating round-off errors.

2. The unfortunate formulation, "many-particle (many-electron) orbitals", has been replaced by many-particle functions, but not everywhere. Following is to be rectified:

- Page 2, the beginning of the section "results": we pre-train the electron `_orbitals_` ... and do not pre-train the positron `_orbitals_`.

- Page 6, middle of column 2: we have included positronic `_orbitals_` as additional species ... (original FermiNet, where spin up and down electronic `_orbitals_` shared weights).

- Page 9, the beginning of "Fermionic neural networks": many-body `_orbitals_ $\psi_i^{k\{x_i\}}$` , which enter into FermiNet determinants.

- Page 10, below eq. 15: The electronic `_orbitals_`, `$\chi_i=(\sigma_i,-)$` , are pre-trained to Hartree-Fock orbitals (the latter are OK, but raising new question - see below).

- Caption for fig. 2: In the benzene molecule, the positronic `_orbital_` is highly diffuse... The positronic

orbital resembles...

This is the positronic density (as the begin of this caption says), well-defined for any wavefunction, and not the positronic orbital! There are no orbitals there!

3. Concerning the process of pre-training the electronic functions, I have new question, to be answered by the Authors. I don't niggle - this question became possible only after clarification that the "orbitals" appear to be many-particle functions. Each function $\psi_i^k(\mathbf{x}_i)$ depends on the coordinates of many particles, and is in reality a correlated function, although not given by an analytical formula. So, how to understand pre-training of such functions to Hartree-Fock orbitals (of bare molecule)? The latter are really one-electron functions.

4. Details.

- Figure 3 is an illustrative example. For which molecule?

- Improved equilibrium bond distance of LiH has been published recently (3.014 bohr, Mol. Phys. e2048107), but 3.015 is OK. Just a notification, without any request to modify the text, or add citations.

- R_{Li_2} is different. Similarly, no request to do more calculations - just provide the source of $R_e=5.015$ bohr for this molecule. NIST reports 5.05 bohr, and so do Nakatsuji and Nakashima (J. Chem. Phys. 157, 093198, 2022).

Reviewer #3 (Remarks to the Author):

I thank the authors for responding to my comments. The paper is high quality and interesting. However, the manuscript still appears to be a benchmarking paper, and there is not a clear set of new results presented in the manuscript.

In my original review I said

"Regardless, the paper does demonstrate an interesting set of technical result for experts in the field. However, there does not appear to be any new physical results in the paper"

The authors' response avoided this issue. There still does not appear to be any new physical results in the revised paper. I am conflicted with the idea of publishing a new method in a high profile journal without there being a great application that can be demonstrated at the time of publication.

While I do appreciate that the authors think this is a step forward for Monte Carlo methods, it remains to be seen if the techniques here will actually gain wide spread usage in comparison to the other techniques in the field. In the conclusion of the current manuscript the authors say

"We expect that the simplicity of this method will lend itself to many challenging applications beyond those presented here, e.g., calculations involving multiple positrons"

I think the challenging applications that the authors mention will indeed be challenging, and such a demonstration certainly would be more of what I would expect. The current manuscript presents a lot of promise, but does not deliver on demonstrating a challenging and new application.

With regards to the rest of the paper and in particular the technical parts, I do think the research is high quality. But most of my comments in my original review still hold. After reading the response from the authors, I still think this work should be published elsewhere. The main results in the current

manuscript are benchmark comparisons to previous techniques, and the expectations that the authors have for this approach to simulate new systems are not realized through actual results presented in the manuscript.

Reviewer #3 (Remarks on code availability):

The code used for the positronic chemistry does not appear to be implemented in the current github repository. The authors suggest they plan to add it at some point in the future.

Reviewer #1:

The authors have revised and improved their manuscript with a more clear description of their objectives, motivations, methods, and new results on annihilation rates. They have shown a detailed explanation for the reviewers' comments. In particular, the accuracy of the annihilation rate which is added in this revision well demonstrates their excellent description of attractive behavior between the electron and positron because the annihilation rate requires higher accuracy to be converged than the energy itself. Notably, the FermiNet-VMC provides the annihilation rate of HPs with higher accuracy (closer to ECG-SVM) than the Fixed Node DMC calculation. The code availability in the public repository will also accelerate the research in positronic chemistry. The authors' replies are considered satisfactory to the reviewers' comments. I would like to comment on the sentence in the Introduction: "The formation of these bound states, enabled by an incident positron exciting a vibrational Feshbach resonance, results in greatly enhanced annihilation rates." Strictly speaking, the vibrational Feshbach resonance states are not bound states that the authors have exactly calculated. The vibrational Feshbach resonance state is surely a quasi-bound state but is decaying into the lower vibrational state + positron.

We thank the referee again for their insightful feedback. Concerning your final comment, we have tweaked the wording of this sentence to provide additional clarity:

*"The formation of these bound states, enabled by an incident positron exciting a vibrational Feshbach resonance **which subsequently decays to the vibrational ground state...**"*

Reviewer #2:

Review of revised article "Neural network variational Monte Carlo for positronic chemistry", by Gino Cassella, W. M. C. Foulkes, David Pfau, and James S. Spencer.

The article, in which the calculations for positronic systems are presented, including non-polar molecules (Li₂, C₆H₆), has been modified accordingly to reviewers' remarks. I accept the arguments of the Authors; in my opinion, most problematic elements of the text are clarified, and the changes head in the right direction. Concerning the non-orthodox presentation order of the scientific material - if this is the Editor's requirement, then I have nothing to say. This paper could be accepted for publication, but after further revision - this time a minor one. My present remarks, to be considered by the Authors, are as follows:

1. I request absolutely the table with single and double precision results, as provided in the rebuttal letter, to be included in the article, with proper comment. I find this information crucial and so could many other readers, that the calculations involving millions (billions?) of summations, carried out on the edge of available precision, are sufficiently numerically stable, and their results do not sink in accumulating round-off errors.

We have now inserted the table provided in our previous response in the Supplementary Material alongside the statement that our calculations were carried out in single precision.

2. The unfortunate formulation, "many-particle (many-electron) orbitals", has been replaced by many-particle functions, but not everywhere. Following is to be rectified:

- Page 2, the beginning of the section "results": we pre-train the electron `_orbitals_` ... and do not pre-train the positron `_orbitals_`.

- Page 6, middle of column 2: we have included positronic `_orbitals_` as additional species ... (original FermiNet, where spin up and down electronic `_orbitals_` shared weights).

- Page 9, the beginning of "Fermionic neural networks": many-body `_orbitals_` $\psi_i^{\mathbf{k}\xi}$, which enter into FermiNet determinants.

- Page 10, below eq. 15: The electronic `_orbitals_`, $\xi=(\sigma,-)$, are pre-trained to Hartree-Fock orbitals (the latter are OK, but raising new question - see below).

- Caption for fig. 2: In the benzene molecule, the positronic `_orbital_` is highly diffuse... The positronic `_orbital_` resembles...

This is the positronic density (as the begin of this caption says), well-defined for any wavefunction, and not the positronic orbital! There are no orbitals there!

We apologise for overlooking these instances of the incorrect phrasing, and thank the referee for diligently locating them on our behalf. Most of these instances have now been amended to read "function" rather than "orbital". In the caption of Figure 2, we have selected an alternative phrasing: "*the positronic **component of the wavefunction** is highly diffuse*", "*the positronic **density** resembles...*".

3. Concerning the process of pre-training the electronic functions, I have new question, to be answered by the Authors. I don't niggle - this question became possible only after clarification that the "orbitals" appear to be many-particle functions. Each function $\psi_i^{\mathbf{k}\xi}$ depends on the coordinates

of many particles, and is in reality a correlated function, although not given by an analytical formula. So, how to understand pre-training of such functions to Hartree-Fock orbitals (of bare molecule)? The latter are really one-electron Functions.

The Hartree-Fock orbitals used in pre-training are indeed one electron functions. In the pretraining procedure, the network is evaluated up to the orbital layer for a Monte Carlo sample, leaving us with a set of N evaluations of the many-electron functions for each of the N electrons for each determinant. The 'loss function' used in pre-training is then the sum of squared differences between these N^2 numbers and the corresponding N^2 numbers obtained by evaluating the N occupied Hartree-Fock orbitals at the positions of the N electrons. This procedure is described in detail in the original FermiNet paper (<https://doi.org/10.1103/PhysRevResearch.2.033429>).

Clearly, pre-training a many-particle function to match a single-particle orbital will not provide any training signal regarding the correlation between electrons. We do not propose a concrete interpretation of the effect this has on the weights of the neural network (although, one suspects, the pretraining has nought but a random effect on the weights corresponding to the features which are pooled over particle streams). Simply, it is an empirical fact that this pretraining procedure results in greater stability (by ensuring initialization of the network to a region of the parameter space where the determinant is non-singular) and faster convergence early in training.

4. Details.

- Figure 3 is an illustrative example. For which molecule?**
- Improved equilibrium bond distance of LiH has been published recently (3.014 bohr, Mol. Phys. e2048107), but 3.015 is OK. Just a notification, without any request to modify the text, or add citations.**
- Li₂ is different. Similarly, no request to do more calculations - just provide the source of R_e=5.015 bohr for this molecule. NIST reports 5.05 bohr, and so do Nakatsuji and Nakashima (J. Chem. Phys. 157, 093198, 2022).**

The results in Figure 3 concern the benzene molecule. The caption has been amended to reflect this.

The bond length of 5.015 Bohr quoted in the manuscript is, fortunately, a simple typo. In fact, the calculations were carried out with the correct bond length of **5.051 bohr**. This bond length was obtained from the NIST Computational Chemistry Comparison and Benchmark DataBase. We thank the referee once again for their diligence in spotting this silly mistake. We have inserted a citation to the source quoted by NIST (Huber and Herzberg 1979).

Reviewer #3:

I thank the authors for responding to my comments. The paper is high quality and interesting. However, the manuscript still appears to be a benchmarking paper, and there is not a clear set of new results presented in the manuscript.

In my original review I said

"Regardless, the paper does demonstrate an interesting set of technical result for experts in the field. However, there does not appear to be any new physical results in the paper"

The authors' response avoided this issue. There still does not appear to be any new physical results in the revised paper. I am conflicted with the idea of publishing a new method in a high profile journal without there being a great application that can be demonstrated at the time of publication.

While I do appreciate that the authors think this is a step forward for Monte Carlo methods, it remains to be seen if the techniques here will actually gain wide spread usage in comparison to the other techniques in the field. In the conclusion of the current manuscript the authors say

"We expect that the simplicity of this method will lend itself to many challenging applications beyond those presented here, e.g., calculations involving multiple positrons"

I think the challenging applications that the authors mention will indeed be challenging, and such a demonstration certainly would be more of what I would expect. The current manuscript presents a lot of promise, but does not deliver on demonstrating a challenging and new application.

With regards to the rest of the paper and in particular the technical parts, I do think the research is high quality. But most of my comments in my original review still hold. After reading the response from the authors, I still think this work should be published elsewhere. The main results in the current manuscript are benchmark comparisons to previous techniques, and the expectations that the authors have for this approach to simulate new systems are not realized through actual results presented in the manuscript.

While we appreciate the generous praise of the quality of our work, we must politely disagree with the referee's evaluation that the results presented are unsuitable for

publication in Nature Communications. From the “Aims & Scope” for Nature Communications (<https://www.nature.com/ncomms/aims>), we quote:

“Nature Communications is an open access, multidisciplinary journal dedicated to publishing high-quality research in all areas of the biological, health, physical, chemical, Earth, social, mathematical, applied, and engineering sciences. Papers published by the journal aim to represent important advances of significance to specialists within each field.”

By the assessment of the referee, our work is high-quality and represents an interesting set of results for specialists within the field. The question remains as to whether this interesting set of results constitutes an ‘important advance’, and to this we answer again in the affirmative. The development of a conceptually simple method with favorable computational scaling, which produces positron binding energies in close agreement with experimental values, and annihilation rates in closer agreement with accurate benchmarks than those obtained by the only other competitive method (in terms of binding energies for large, non-polar molecules) can only be considered to be an important advance by specialists within the field.

Additionally, we would like to highlight some examples of manuscripts published in Nature Communications which do not present ‘new physical results’, but nonetheless offer significant advances through methods development. From our own field of neural network variational Monte Carlo we highlight the work of Entwistle *et al.* regarding the computation of excited states: <https://www.nature.com/articles/s41467-022-35534-5>, the work of Ren *et al.* regarding the application of neural network trial wavefunctions in diffusion Monte Carlo calculations: <https://www.nature.com/articles/s41467-023-37609-3>, and the work of Choo *et al.* regarding the development of neural network quantum states in second quantization applied to the calculation of ground-state energies of small molecules: <https://www.nature.com/articles/s41467-020-15724-9>. Doubtless, there are numerous similar examples from other subfields of computational chemistry and physics.

We would like to once again thank the referee for their comments.

Reviewer #3 (Remarks on code availability):

The code used for the positronic chemistry does not appear to be implemented in the current github repository. The authors suggest they plan to add it at some point in the future.

We are in the process of preparing the code for public release in the open-source FermiNet repository, and expect to be able to do so soon if the manuscript is accepted for publication.

Reviewer #2 (Remarks to the Author):

Review of 2'nd revised article "Neural network variational Monte Carlo for positronic chemistry", by Gino Cassella, W. M. C. Foulkes, David Pfau, and James S. Spencer.

The article, which presents the machine learning approach to the challenging problem of building correlated wavefunctions for variational Monte Carlo calculations for positronic molecules, has been amended to the point where I can recommend it for publication in Nature Communications.

I have no further requests. It is clearly demonstrated that the method is capable to provide competitive results for molecules as large as positronic benzene, and the errors of binding energies of small systems (PsH, e+LiH) are within a fraction of milihartree. Apart of some molecules whose ability to bind positron was already known, the lithium molecule (Li₂ - I don't know why is the strange name of "dilithium" enforced in chemistry, but I feel that the responsible persons are certainly "Star Trek" fans) with attached positron is studied for the first time. The calculations of the positron affinity of this molecule and of the annihilation rate of e⁺+Li₂ are an important element of scientific novelty in the work under review - even if positronic Li₂ is not necessarily an attractive subject for today's experiments. I see it could be a possible future benchmark system for testing computational methods, as the simplest positronic non-polar molecule. Bravo!

We have no additional comments, except to thank the Referee for their kind words